# Enhanced biennial variability in the Pacific due to Atlantic capacitor effect

Lei Wang[1,2], Jin-Yi Yu[1] & Houk Paek[1]

The El Niño-Southern Oscillation (ENSO) and the variability in the Pacific subtropical highs (PSHs) have major impacts on social and ecological systems. Here we present an Atlantic capacitor effect mechanism to suggest that the Atlantic is a key pacemaker of the biennial variability in the Pacific including that in ENSO and the PSHs during recent decades. The 'charging' (that is, ENSO imprinting the North Tropical Atlantic (NTA) sea surface temperature (SST) via an atmospheric bridge mechanism) and 'discharging' (that is, the NTA SST triggering the following ENSO via a subtropical teleconnection mechanism) processes alternate, generating the biennial rhythmic changes in the Pacific. Since the early 1990s, a warmer Atlantic due to the positive phase of Atlantic multidecadal oscillation and global warming trend has provided more favourable background state for the Atlantic capacitor effect, giving rise to enhanced biennial variability in the Pacific that may increase the occurrence frequency of severe natural hazard events.

[1] Department of Earth System Science, University of California, Irvine, California 92697, USA. [2] Guangdong Province Key Laboratory for Coastal Ocean Variation and Disaster Prediction, Department of Oceanography, College of Ocean and Meteorology, Guangdong Ocean University, Zhanjiang 524088, China. Correspondence and requests for materials should be addressed to L.W. (email: leiwangocean@yahoo.com).

The El Niño-Southern Oscillation (ENSO) and Pacific subtropical highs (PSHs) affect the livelihoods of people worldwide through their influences on severe natural hazards including tropical storms, coastal erosion, droughts and floods[1-3]. The ability to forecast ENSO and PSH variability requires an understanding of the underlying physical mechanisms that drive their fluctuations. ENSO is known to have two dominant frequency bands: the quasi-biennial (QB) band with a period of 2–3 years and the low-frequency (LF) band with a typical period of 3–7 years[4,5]. Variations in ENSO can be described as combinations of the varying phases and amplitudes of these two components. Interdecadal changes can be observed in the amplitudes of these two bands, giving rise to the interdecadal changes in ENSO frequency[6,7]. The QB component is considered to be one major mechanism responsible for the spring persistence barrier of ENSO[8], which limits how far ahead that we can predict ENSO events. No consensus has been reached on the root cause of the QB variability in ENSO. Most previous studies have emphasized processes within the tropical Pacific or Indian Oceans for the generation of the QB ENSO[9-16], pointing the pacemaker to be mainly lied within the Indo-Pacific Oceans.

The Atlantic has not been considered to be a key driver of the QB ENSO. However, recent studies have increasingly suggested that the Atlantic Ocean plays an active role in forcing Pacific climate and ENSO variability[17-25]. In particular, Ham et al.[20] found that sea surface temperature (SST) anomalies in the North Tropical Atlantic (NTA) during boreal spring can trigger ENSO events in the following winter. This Atlantic-induced ENSO is characterized by SST anomalies that are strongest in the tropical central Pacific, and resembling the so-called Central Pacific (CP) type of ENSO but not the traditional Eastern Pacific (EP) type[26,27]. Most previous studies of the QB ENSO have not differentiated these two types of ENSO and have implicitly focused more on the QB component of the conventional EP ENSO. The centre of warm SST anomalies associated with El Niño has been found to move from the eastern Pacific to central Pacific during recent decades[28], which was suggested to be a Pacific Ocean response to a phase change in the Atlantic multidecadal oscillation (AMO)[24]. Moreover, the CP ENSO has a stronger QB component than the EP ENSO[27]. Thus, it is possible that the Pacific–Atlantic interaction or coupling may have become more important to ENSO variability, particularly the QB component, as ENSO changed from the EP type to the CP type during recent decades.

Here we perform statistical analyses and numerical experiments to show that the Atlantic Ocean is a key pacemaker of the biennial variability in Pacific, including that in ENSO and the PSHs during the recent two decades. An Atlantic capacitor effect mechanism is identified to suggest that the Pacific–Atlantic interactions have been stronger since the early 1990s, enhancing the biennial variability in the Pacific. Since the early 1990s, a warmer Atlantic due to the positive phase of the AMO, and global warming trend has provided a more favourable background state for the Atlantic capacitor effect to feedback from the Atlantic to the Pacific. Our results highlight the increasing influences of the Atlantic Ocean on Pacific climate and variability in recent decades.

## Results

**Correlations between ENSO and NTA SST indices.** To elucidate the possible role of the Pacific–Atlantic interaction, we first performed a 21-year sliding correlation analysis between the Niño3.4 index during boreal winter (December–January–February; DJF) and the NTA SST index (Method) during the

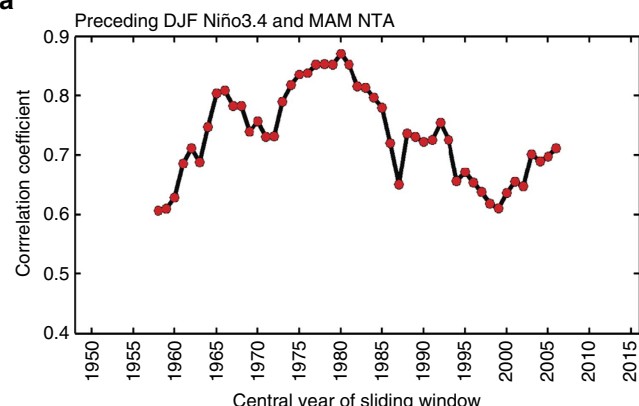

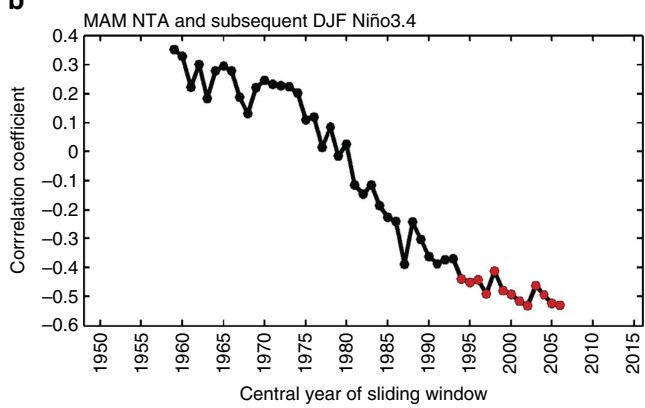

**Figure 1 | Correlations between the ENSO and NTA SST indices.** (**a**) The 21-year sliding correlation coefficients (for example, correlation coefficient in 2002 representing 1992–2012) between preceding boreal winter (DJF) Niño3.4 and spring (MAM) NTA SST during 1948–2016. (**b**) Same as **a** but for MAM NTA SST and subsequent DJF Niño3.4. Following ref. 20, the impact of the previous winter ENSO has been excluded in **b** by removing the linear regression with respect to the Niño3.4 index during the previous DJF season. The linear trend has been removed within each sliding window for both indices before calculating the sliding correlation coefficients. The red dots represent correlations that are significant at the 95% confidence level.

subsequent spring (March–April–May; MAM) for the period 1948–2016. The DJF value of Niño3.4 index is selected to represent the ENSO intensity during its typical peak season, while MAM is the season in which the NTA SST anomalies show the strongest correlation with the ENSO SST anomalies (Supplementary Fig. 1). We find the correlation coefficients to be positive and significant throughout the analysis period (Fig. 1a), which indicates that winter tropical Pacific warming (cooling) associated with ENSO can induce warming (cooling) in the NTA SST during the subsequent spring. This imprint process can be considered as a charging process to the tropical Atlantic, which is analogous to the charging of a capacitor[29]. We next examine whether the spring NTA SST can also exert an influence on ENSO during the subsequent winter, as in a discharging capacitor. We find the 21-year sliding correlations between the spring NTA SST and the following winter Niño3.4 index are statistically significant only after the early 1990s (Fig. 1b). The negative correlations indicate that a NTA warming in spring is followed by a Niño3.4 cooling in the following winter and vice versa.

The sliding correlation analysis indicates that, after early 1990s, an El Niño condition in the tropical Pacific can induce a NTA

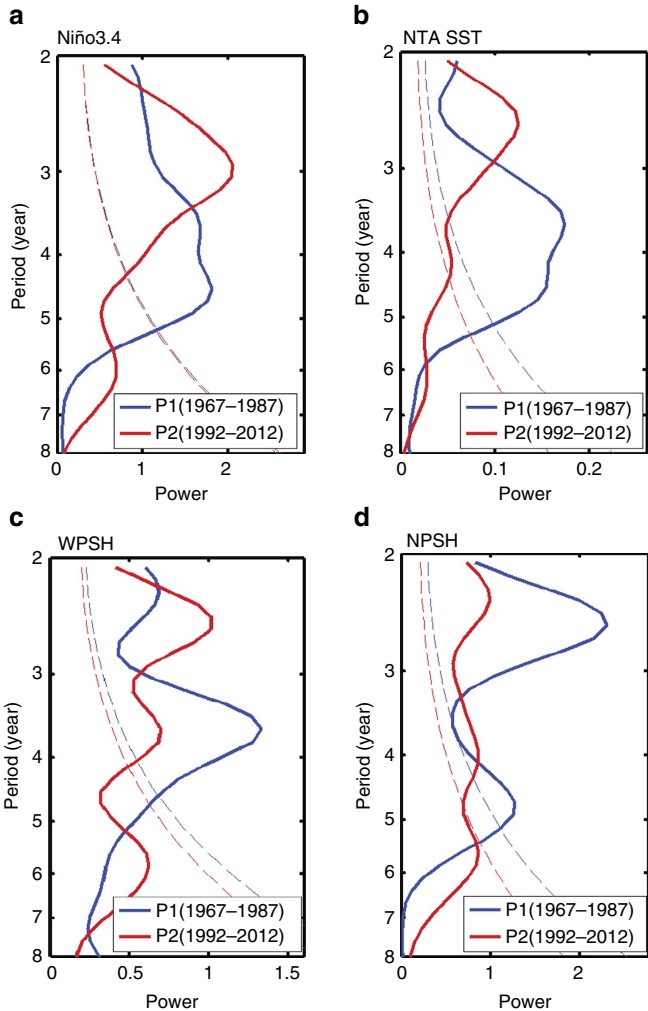

**Figure 2 | Comparison of power spectrum for climatic indices before and after the early 1990s.** (a–d) Distribution of power spectrum for DJF Niño3.4 index (**a**), MAM NTA SST (**b**), JJA WPSH index (**c**) and JJA NPSH index (**d**). The dashed line denotes the 95% confidence interval against red noise.

warming in the following spring, which then feedbacks to trigger a La Niña condition in the Pacific during the second winter. Similarly, a La Niña in year one can be followed by an El Niño condition in year two with the help of the feedback from the La Niña-induced cooling in the NTA region. This Atlantic feedback serves as a phase-reversal mechanism for the ENSO cycle leading to changes in-phase every 2 years, giving rise to a QB component of ENSO. Since the Atlantic feedback is significant only after early 1990s, we should expect an intensification of the QB component of ENSO after that time. We examine the power spectrum of winter Niño3.4 index (Fig. 2a) and find the index to be dominated by a LF frequency during the 21-year period 1967–1987 (P1 hereafter) before the early 1990s but by the QB frequency during the 21-year period 1992–2012 (P2 hereafter) afterward. A similar periodicity change is also found for the spring NTA SST (Fig. 2b).

As mentioned, influences from the Indian Ocean can also contribute to the QB tendency of the ENSO. To exclude Indian Ocean influences, we repeat the analysis but removing the regression onto the Indian Ocean Basin mode index (Method) and obtain similar results (Supplementary Fig. 2). This result suggests that the increased QB tendency of ENSO and the increased Atlantic–Pacific correlation after the early 1990s are probably not caused by influences from the Indian Ocean.

**Pacific–Atlantic interactions and biennial variability.** It is necessary to understand why ENSO can always induce NTA SST anomalies but the NTA SST anomalies feedback to ENSO only after the early 1990s. ENSO is known to be able to influence the other ocean basins, including the Atlantic, through the atmospheric bridge mechanism[30]. There are two possible atmospheric 'bridges' that can link ENSO to Atlantic SST variations: one is the Pacific North American (PNA) teleconnection pattern[30,31], and the other is the anomalous Walker and Atlantic Hadley circulations[32]. Both atmospheric 'bridges' can modulate the strength of the North Atlantic subtropical high (NASH) and its associated trade winds (and therefore surface evaporation) to induce NTA warming or cooling. NTA SST anomalies induced by ENSO can then persist from the winter season when ENSO typically reaches its peak intensity through the subsequent spring via local air–sea interactions. Correlation analyses (Supplementary Table 1) confirm this sequence of processes by showing significant correlation coefficients (at the 95% confidence interval) among winter ENSO index, indices of the atmospheric bridges (that is, the PNA and Atlantic Hadley circulation), and NTA SST index (Methods). The correlations are significant during both P1 and P2, except that the PNA atmospheric bridge mechanism is found only during P2 while the Hadley circulation bridge mechanism is manifested during both periods.

The spring SST anomalies in the NTA have been verified to influence ENSO by affecting the subtropical teleconnections along the Intertropical Convergence Zone (ITCZ), which relays the Atlantic signal to the Pacific[20]. The inter-basin interaction mechanism of Ham et al.[20] suggests that the positive NTA SST anomalies intensify the Atlantic ITCZ to excite a Gill-type response[33] over the Eastern Pacific, which then triggers subtropical Pacific air–sea-coupled interactions to give rise to negative SST anomalies in the tropical central Pacific a few months later. In this study, we show that this mechanism is at work only after the early 1990s but not before. A regression analysis (Fig. 3) shows that the spring NTA warming can induce an equatorial Pacific cooling in the subsequent winter during P2 but not during P1. The wind vectors in Fig. 3 indicate that NTA warm anomalies excite a pair of low-level circulation responses along the ITCZ to relay the Atlantic influence into the Pacific. The induced Pacific SST anomalies closely resemble those associated with the CP ENSO[24,27], whose SST anomalies typically appear first off the Baja California coast, spread southwestward toward the equator, and peak in the equatorial central Pacific. The QB ENSO induced by this Atlantic feedback is apparently of the CP type. It should be noted that Atlantic SST anomalies can also impact Pacific SST anomalies in the following seasons by modulating the Walker circulation[18,19], but this mechanism primarily involves SST anomalies in the equatorial Atlantic (such as those associated with Atlantic Niño events) but not the anomalies in the NTA region.

It is interesting to note from Fig. 3e that the anticyclonic anomalies associated with the NTA-induced subtropical teleconnection are co-located with the two major components of the summer Pacific high, which are often referred to as the western Pacific subtropical high (WPSH) and north Pacific subtropical high (NPSH). A regression analysis between the summer (June–July–August; JJA) sea level pressure (SLP) anomalies and the spring NTA SST (Supplementary Fig. 3) confirms that the spring NTA SST anomalies became more capable of influencing the intensities of these two PSHs after the early 1990s. The subtropical teleconnection associated with the spring NTA SST anomalies tends to induce an in-phase relationship between the WPSH and NPSH variations, which is consistent with the recent

                                                                 **3**

**Figure 3 | Regression with respect to spring NTA SST.** (**a–c**) Regressions of SSTAs (shading) and 850-hPa winds (vector) with respect to the MAM NTA SST during P1 for MAM (**a**), JJA (**b**) and DJF (**c**) seasons. (**d–f**) Same as **a–c** but for during P2 for MAM (**d**), JJA (**e**) and DJF (**f**) seasons. The impact of the previous winter ENSO has been excluded by removing the linear regression with respect to the Niño3.4 index during the previous DJF season. Only the values at the 90% confidence level or higher are shown.

findings of Paek *et al.*[34]. Yun *et al.*[35] also suggested that the in-phase covariability of the WPSH and NPSH during boreal summer contributes to the QB component of ENSO. Here, we demonstrated that the spring NTA SST can contribute to the in-phase pattern of the WPSH and NPSH after the early 1990s, via subtropical teleconnections along the ITCZ and by exerting mutual influences on the WPSH and NPSH. Therefore, the summer WPSH and NPSH are both involved in the Atlantic–Pacific interactions that intensify the QB component of the ENSO after early 1990s. The leading periodicities of the summer WPSH index also shifted from the LF to the QB (Fig. 2c) in early 1990s, which was also noted in a previous study[36]. For the NPSH, significant QB periodicity is also observed after early 1990s (Fig. 2d). However, the dominant periodicity for the NPSH before early 1990s was already concentrated in the QB band, which seems to have no close association with the NTA SST and may be influenced by other factors. The stronger Atlantic influence after the early 1990s has intensified the QB component in both ENSO and the WPSH, both of which strongly modulate extreme weather activities in the Asian-Pacific region.

Through two-way feedbacks between the Pacific and Atlantic with alternating 'charging' and 'discharging' processes, rhythmic changes of ∼ 2-year periodicity have been intensified in the Pacific after early 1990s including the ENSO and variability in the PSHs (Fig. 4). We find this sequence of Atlantic–Pacific coupling and the biennial cycle especially prominent during the 2009/10 El Niño event (Supplementary Fig. 4). NTA cooling in spring 2009 induced a CP El Niño in the winter of 2009. This El Niño further

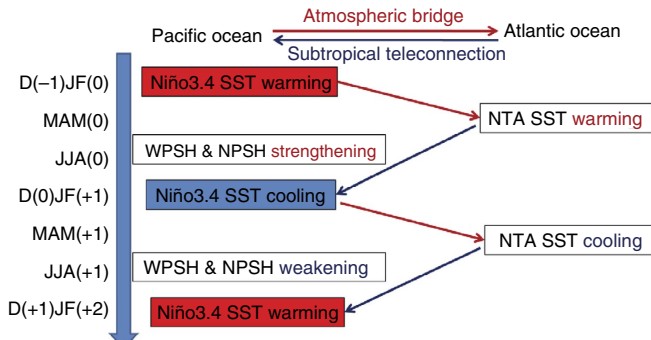

**Figure 4 | A schematic diagram showing the biennial cycle of the ENSO induced by the Atlantic capacitor effect.** Prominent biennial variability in the boreal winter Niño3.4, spring NTA SST, summer WPSH and NPSH existed in this Atlantic–Pacific-coupled process. The red arrow represents the 'charging' process in which the preceding winter ENSO influences the spring NTA SST like a battery charges a capacitor via an atmospheric bridge mechanism, and the blue arrow is for the 'discharging' process in which the NTA SST exerts its climatic influences on the following ENSO like a discharging capacitor via a subtropical teleconnection mechanism.

induced anomalous NTA warming in the subsequent spring that feedback to the Pacific to induce the 2010 La Niña event. The QB cycle resulting from the 2009 El Niño to 2010 La Niña fits nicely the Atlantic capacitor effect that we depict in this study.

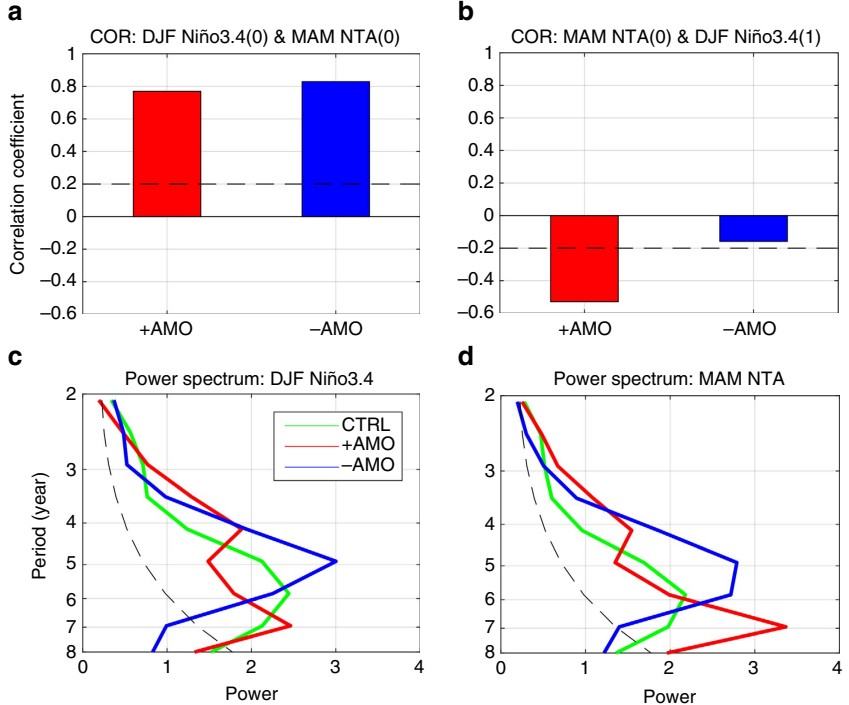

**Figure 5 | Changes in the effectiveness of the Atlantic capacitor effect in numerical experiments with different AMO phases.** (**a**) The 'charging' process represented by the correlation coefficients between the winter Niño3.4 index and the following spring NTA index in the positive-AMO run and the negative-AMO run. The dashed line denotes the 95% confidence interval. (**b**) The 'discharging' process represented by the correlation coefficients between the spring NTA index and the subsequent winter Niño3.4 index. The previous winter Niño3.4 regressions had been removed from NTA index before calculating coefficient. (**c**) Power spectra of the winter Niño3.4 indices in CTRL, positive and negative-AMO runs. The dashed line denotes the 95% confidence interval against red noise. (**d**) As in **c** but for the spring NTA indices.

**Influences from a warmer Atlantic after the early 1990s.** Changes in the background states of the Atlantic or/and Pacific could be the reason why the Atlantic is more capable of influencing the Pacific after the early 1990s than before. We examine the mean state differences between P1 and P2 (Supplementary Fig. 5) and find the spring NTA SST to be significantly warmer during P2 (Supplementary Fig. 5a). Accompanying the warmer SSTs, were enhanced convection and precipitation (Supplementary Fig. 5b,c) along the ITCZ that extends from the Atlantic to the eastern Pacific. The stronger ITCZ after the early 1990s is more likely capable of channelling the Atlantic influence into the eastern subtropical Pacific. The warmer basic state in the NTA region after the early 1990s is likely related to the phase change of the AMO and global warming trends. The possible interference of a global warming trend may be a reason why the previous positive phase of the AMO (during the 1930s–1960s) was not accompanied by enhanced Atlantic–Pacific interactions. As shown in Fig. 1b, the correlation between MAM NTS index and the following DJF Nino3.4 index is significant and negative during the recent positive-AMO period (after the early 1990s), but weak and not significant during the previous positive-AMO phase (before the 1960s). Prominent multi-decadal variability can be identified in the time series of the NTA SST index after removing the linear trend (Supplementary Fig. 6c). The multi-decadal variability is closely associated with the AMO index, both of which switch sign in the early-to-middle 1990s. It is noted that the correlation coefficient between NTA and equatorial Pacific SST anomalies has changed from insignificant positive values in the 1950s to significant negative values after the early 1990s. This decreasing trend of the sliding correlations may be related to the global warming trend (Supplementary Fig. 6a). The positive phase of the AMO after the early 1990s added an additional warming in

the Atlantic Ocean to boost the correlations above the 95% significance level.

Three sets of ensemble experiments were further conducted with a coupled atmosphere-ocean model (Method) to demonstrate that the effectiveness of the Atlantic capacitor effect can indeed change with the background SSTs in the tropical Atlantic Ocean during the positive and negative-AMO phases. In the positive-AMO ensemble run, a significant positive correlation was produced between the preceding DJF Niño3.4 and MAM NTA SST that represents the 'charging' process of the capacitor effect, while a significant negative correlation was produced between MAM NTA SST and the following DJF Niño3.4 (Fig. 5a,b) that represents the 'discharging' process of the capacitor effect. In contrast, in the negative-AMO ensemble run, only the significant positive correlation was produced but not the significant negative correlation. These modelling results confirm the observational finding that the feedback from the Atlantic NTA region to the Pacific ENSO is stronger in the positive phase of the AMO than in the negative phase of the AMO. We also examine the ENSO periodicity in the model experiments (Fig. 5c). The CTRL run of the NCAR CESM model is dominated by a 6-year ENSO. The biennial periodicity of ENSO simulated in the model is very weak. The CESM model ENSO is likely dictated by the Pacific ocean-atmosphere coupling processes that give rise to a LF (that is, 4–6 years) periodicity. In the positive-AMO run, the Atlantic capacitor effect provides an additional coupling mechanism that tends to give rise to a QB ENSO periodicity. A nonlinear combination of the Pacific coupling mechanism and the Atlantic capacitor effect can give rise to ENSO periodicities near 4 years [(6 year + 2 year)/2 = 4 year]. It is encouraging to see that the dominant ENSO period in the positive-AMO run splits into two, one centred close to 6 years and the other centred at 4 years. No

such splitting of ENSO periodicity was found in the negative-AMO run, where the ENSO periodicity is still dominated by a single peak around 5 years, consistent with the lack of the Atlantic capacitor effect. These modelling experiments support our observational findings that the Atlantic capacitor effect can work more efficiently during the positive-AMO phase, which then contributes to enhance biennial variability in ENSO.

Our results suggest that the increased feedback from the Atlantic to the Pacific has enabled an Atlantic capacitor effect to intensify the biennial variability in the Pacific after early 1990s. Our suggestion is very different from previous prevailing ideas that have emphasized the Indo-Pacific Oceans as the pacemaker for the biennial variability in ENSO[8–16]. Our study also invokes the influence from the Atlantic Ocean to explain why ENSO has changed from the EP type to the CP type during recent decades. In association with the increased biennial variability, extreme weather and climate events in the Pacific, such as El Niño/La Niña and drought and flood events associated with typhoons or the PSH, may occur more frequently in the near future.

## Methods

**Observations and reanalysis data.** The monthly SST data used is from NOAA's Extended Reconstructed Sea Surface Temperature version 3b (ERSST)[37] with a $2° × 2°$ horizontal resolution. Monthly tropospheric winds and SLP data sets beginning in 1948 are from the National Centers for Environmental Prediction-National Center for Atmospheric Research (NCEP-NCAR) reanalysis[38] with a $2.5° × 2.5°$ resolution. The NOAA outgoing long-wave radiation (OLR) data[39] with a resolution of $2.5° × 2.5°$ that begins in June 1974 is used. The Global Precipitation Climatology Project (GPCP)[40] monthly precipitation analysis derived from satellite and gauge measurements that starts in 1979 is used. All the anomalies are calculated by first removing the long-term trend and then removing the mean seasonal cycle based on the period 1971–2000.

**The model simulations.** The NCAR CESM 1.1.2 coupled atmosphere-ocean general circulation model (CGCM) was used for the modelling experiments. Three CGCM experiments were conducted: a control (CTRL) run, a positive-AMO run and a negative-AMO run. For the AMO runs, surface heat flux anomalies (that is, the AMO-regressed SST anomalies multiplied by a factor of 100) are prescribed over the North Atlantic to force the model to produce positive or negative-AMO anomalies. This modelling methodology is adapted from Zhang and Delworth[41]. This prescribed heat flux was added only to force the oceanic model component and thus does not interfere with the surface heat flux anomalies that drive the atmospheric response to the NTA SST anomalies. The simulated AMO index is $+0.4 °C$ in the positive-AMO run and $−0.4 °C$ in the negative run (compared with that in the CTRL run), which is about twice the observed amplitude of the AMO index[42]. A total of 100 years of integrations from the CTRL run and 96 years from each of the two AMO runs were used in the analysis.

**Climate indices.** The NTA SST index is defined as the SST anomalies averaged over $[0°–15° N, 80° W–20° E]$. The Niño3.4 index, used to represent ENSO intensity, is defined as the SST anomalies averaged over $[5° S–5° N, 170° W–120° W]$. The Indian Ocean Basin index is defined as the SST anomalies averaged over $[20° N–20° S, 40° E–110° E]$, to represent the Indian Ocean warming. The AMO index[43] is calculated as the detrended SST anomalies averaged over the North Atlantic from the equator to the 70° N. Following ref. 35, the WPSH and NPSH indices were calculated as the SLP anomalies averaged over $[15° N–25° N, 110° E–150° E]$ and $[30° N–40° N, 170° W–140° W]$, respectively. The PNA index[44] was constructed based on the 500 hPa geopotential height anomalies. Following the method defined by Schwendike et al.[45], the Atlantic Hadley circulation index was defined as the 500 hPa vertical mass flux difference between an equatorial $[5° S–5° N, 80° W–20° W]$ and a subtropical band $[10° N–25° N, 80° W–20° W]$.

**Significance tests.** We determined the statistical significance levels based on the two-tailed $P$-values using a Student's $t$-test. The number of effective degrees of freedom was determined by using the methods introduced by Bretherton et al.[46].

**The selection of the pre- and post-early 1990s period.** In the analysis, P1 (1967–1987) and P2 (1992–2012) were selected based on the 21-year sliding correlations (Fig. 1b), which show that the maximum absolute value of the correlation coefficient ($−0.53$) occurred in 2002 and the minimum absolute value ($0.01$) occurred in 1977. A 21-year period centring at 1977 (that is, P1) is defined to represent the pre-early 1990s period, while a 21-year period centring in 2002 (that is, P2) is defined to represent the post-early 1990s period.

**Data availability.** The ERSST data set is available at http://www1.ncdc.noaa.gov/pub/data/cmb/ersst/v3b/netcdf/. The NCEP/-NCAR monthly reanalysis is available at http://www.esrl.noaa.gov/psd/data/gridded/data.ncep.reanalysis.html. The OLR and GPCP data sets are provided by the NOAA/OAR/ESRL Physical Sciences Division, from their Web site at http://www.esrl.noaa.gov/psd/. The AMO index is obtained from the Physical Sciences Division at http://www.esrl.noaa.gov/psd/data/timeseries/AMO/. The monthly PNA index is provided by the Climate Prediction Center at http://www.cpc.ncep.noaa.gov/products/precip/CWlink/pna/month_pna_index2.shtml.

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

## Acknowledgements

This work was supported by the National Science Foundation's Climate and Large Scale Dynamics Program under Grant AGS-1505145. We would like to acknowledge high-performance computing support from Yellow stone provided by NCAR's Computational and Information Systems Laboratory, sponsored by the National Science Foundation. L.W. thanks the China Scholarship Council and the Guangdong Natural Science Foundation under Grant 2015A030313796 for supporting his visit to University of California, Irvine. We thank three anonymous reviewers for constructive comments on previous drafts of this manuscript.

## Author contributions

L.W. and J.-Y.Y. designed the study and wrote the paper. L.W. performed the analyses. H.P. conducted the modelling experiments and contributed to the interpretation of the results.

## Additional information

**Competing interests:** The authors declare no competing financial interests.

**Publisher's note**: 

