## [Peer Review File · Nature Communications]

Reviewers' comments:

Reviewer #1 (Remarks to the Author):

The paper by Wang et al. investigates the mechanism by which the tropical Pacific and the tropical Atlantic climate are coupled on interannual timescales. The authors present two main conclusions. First, they find enhanced biennial variability in the two ocean basins and the overlying atmosphere since 1990. Second, the enhanced biennial component is attributed to an Atlantic capacitor effect, where the coupling between the tropical Pacific and the tropical Atlantic is through an atmospheric bridge. I find the story quite convincing and recommend publication subject to only minor revisions.

Minor points:

1. Can the authors speculate about the mechanism for the positive correlation between MAM NTA and DJF Niño3.4 seen in Fig. 1b? Although the correlation is not significant at the 95% level, it may be significant at the 90% level during the late 1950s and during the 1960s.
2. The coupling mechanism during JJA and thereafter looks pretty much like happening through the Walker Circulation, as suggested in Rodríguez-Fonseca et al. (2009) and Ding et al. (2011).
3. Change the color bar in Suppl. Fig. 2 for OLR and precipitation. Enhanced rainfall should be shown in blue.

Reviewer #2 (Remarks to the Author):

This study presents an Atlantic capacitor effect mechanism to enhance the biennial variability of the ENSO. The proposed mechanism can be summarized as follows; The El Niño tends to warm the SST over the NTA region with several months lag about one season, then, the NTA warming during boreal spring season can lead the La Niña during the subsequent winter season. This mechanism is enhanced especially during recent decades with aids of the positive phase of the AMO and global warming trend. The proposed mechanism is plausible and convincing, however, in my opinion, it is not fully validated. One of my main concerns is that the most of results in this study can be shown even though biennial tendency of the ENSO can be excited by other factors except for the NTA. For example, both the biennial tendency of the NTA SST and ENSO would be enhanced if the biennial tendency of the ENSO can be led by the Indian Ocean-ENSO coupling. To address this concern, authors should perform the additional analysis like partial regression to exclude the factors that can lead the biennial tendency of the ENSO, and the model experiments can be another option to clarify their mechanism. Therefore, I recommend this article to be published after the major revisions.

Major Comments:

1. Authors argue that the NTA can play a critical role to the biennial tendency of the ENSO after early 1990s. The mechanism is promising, but I feel the supporting materials are not clear. For example, as the Indian Ocean warming can lead the biennial tendency of the ENSO as authors mentioned in the introduction, it is worthwhile to check whether their relationship is still valid after removing the impact of the Indian Ocean warming. Also, the foot-printing mechanism that infers the interaction between the tropical and mid-latitude Pacific is worthwhile to be removed.

In addition, authors show their proposed mechanism is valid for individual cases in more detail as shown in Supplementary Fig. 4.

2. The coupled model experiments with and without the role of the NTA SST variability are worthwhile to be considered. As mentioned in the comment #1, the analysis using the observational data has a limitation to exclude various factors to lead the biennial tendency of the ENSO. The model experiment clarifies their mechanism is critical to the enhanced biennial tendency during recent decades. Or, at least authors should analyze the long-term integration of

the coupled models whether their mechanism is supported in the modeling framework.

3. Authors show using the moving correlation that the inverse relationship between the NTA warming and the subsequent ENSO is gradually increasing during recent decades. They suspected the climatological SST warming over the Atlantic related to the AMO and the global warming is responsible for this enhanced relationship. However, the SST related to the AMO does not show any linear-trend, which might imply that the global warming might strongly affect to the enhanced NTA-subsequent ENSO relationship during recent decade. Authors need to clarify how the AMO can play a role to the enhanced NTA-subsequent ENSO relationship.

Minor comments:

1. page3 : There are many previous literatures to examine the inter-decadal variation of the ENSO frequency (An and Wang, 2000; Wang et al., 2008).

2. page3, line87: There is another paper from Ham et al. (2013) about the role of the NTA warming.

3. page157-178: The linkage between the Atlantic and the Pacific through two summer Pacific high-pressure system (WPSH, and NPSH) needs to be elaborated for the easy understanding of the proposed mechanism. Does the weak biennial tendency of the NPSH in recent decades imply that the role of NPSH is not critical on the proposed mechanism?

References

An, S. I., & Wang, B. (2000). Interdecadal change of the structure of the ENSO Mode and its impact on the ENSO frequency*. *Journal of Climate*, 13(12), 2044-2055.

Wang, B., Yang, J., Zhou, T., & Wang, B. (2008). Interdecadal changes in the major modes of Asian-Australian monsoon variability: strengthening relationship with ENSO since the late 1970s*. *Journal of Climate*, 21(8), 1771-1789.

Ham, Y. G., Kug, J. S., & Park, J. Y. (2013). Two distinct roles of Atlantic SSTs in ENSO variability: north tropical Atlantic SST and Atlantic Niño. *Geophysical Research Letters*, 40(15), 4012-4017.

Reviewer #3 (Remarks to the Author):

In this manuscript, the authors investigated the atmospheric bridges between the tropical Pacific and the tropical Atlantic using statistical analyses. They showed that the North Tropical Atlantic plays an important role in the transition between El Niño and La Niña. This mediating effect becomes more important since the early 1990s. I think this work is very interesting. In particular, Fig1b shows a beautiful relationship, which may contribute to the predictability of El Niño events. Nonetheless, I do have several concerns and suggestions especially about the evidences of this study, the details of which are listed as follows:

1. Figure 1b is the key figure of this paper (it shows a very beautiful result, indicating that the role of Atlantic has changed during the past 50 years). Here I suggest the authors better clarify their method: How did they calculate the sliding correlation coefficients and the significance level? Whether or not did the authors remove the linear trend within each sliding window? How did the authors estimate the degree of freedom when estimating the significance level (Considering that the 95% confidence level is a constant, I assume the authors fix the degree of freedom according to the sample size, however here I suggest them to follow the methods introduced by Bretherton et al 1999). Clarification of the method may help strengthen their most important result.

2. Similarly, in Figure3, did the authors remove the linear trend of each period before performing the regression analyses? Please clarify.

3. In this paper, the authors did not provide enough evidence to show the causality and the mechanism of the teleconnection between the Atlantic and the Pacific Ocean. I suggest the authors perform numerical experiments to directly show that the Atlantic "capacitor" effect can indeed change according to the background temperature of the tropical Atlantic Ocean.

4. Another question that I have is why the authors chose MAM NTA temperature as the key indicator (e.g. in Figure1 and Figure2)? I understand that the MAM NTA SST may have the strongest correlation with the preceding DJF Nino SST. However, it will be better for the authors if the rationale is further explained and this relationship is further clarified.

In addition, the correlation between the tropical Atlantic SST and the subsequent Nino 3.4 SST may depend on the seasons selected (of both Atlantic SST and Nino SST). Some additional statistically analysis would be nice to show why MAM NTA SST has been selected as the indicator

5. Another concern that I have is whether the Pacific climate becomes more sensitive to the Atlantic SST since 1990s, or it is the warming trend in Atlantic that favors a La Niña mode in the Pacific Ocean, considering that there is a long term warming trend over the tropical Atlantic Ocean? This is another reason that I suggest the authors to perform some numerical simulations to directly show the causality of this relationship.

Reference

Bretherton, C. S., Widmann, M., Dymnikov, V. P., Wallace, J. M. & Blade', I. The effective number of spatial degrees of freedom of a time-varying field. *J. Clim.* 12, 1990–2009 (1999).

Point-by-Point Replies to Reviewer #1

We thank the reviewer for the constructive and helpful comments. Our point-by-point replies are as follows: (The line numbers mentioned here are based on the track-change version of the manuscript)

=====

The paper by Wang et al. investigates the mechanism by which the tropical Pacific and the tropical Atlantic climate are coupled on interannual timescales. The authors present two main conclusions. First, they find enhanced biennial variability in the two ocean basins and the overlying atmosphere since 1990. Second, the enhanced biennial component is attributed to an Atlantic capacitor effect, where the coupling between the tropical Pacific and the tropical Atlantic is through an atmospheric bridge. I find the story quite convincing and recommend publication subject to only minor revisions.

Minor points:

1. Can the authors speculate about the mechanism for the positive correlation between MAM NTA and DJF Niño3.4 seen in Fig. 1b? Although the correlation is not significant at the 95% level, it may be significant at the 90% level during the late 1950s and during the 1960s.

REPLY: Firstly, we would like to point out that this positive correlation cannot contribute to the biennial variability in the ENSO, which is the main focus of this study. Thus, this positive correlation does not affect the main results and conclusions of this study on biennial ENSO tendency.

We speculate that other factors, such as the JJA Atlantic Niño, may play some role in the positive correlation during 1950s and 1960s. It is known that equatorial Atlantic SST anomalies (i.e., the Atlantic Niño events) during boreal summer (June-July-August, JJA) have a negative correlation with eastern-to-central Pacific SST anomalies (Rodríguez-Fonseca et al. 2009 and Ding et al. 2002).

Figure A1. The 21-year sliding correlation coefficients between the MAM NTA SST and JJA Atl3 index. The correlation coefficients are not significant at the 95% level.

We find the MAM NTA SST tend to be negatively correlated with the subsequent JJA Atl3 index (3°S–3°N; 20°W–0°) during 1950s and 60s (see Fig. A1). It is possible that warm (cold) MAM NTA SST anomalies during these two decades induced cold (warm) SST anomalies in Atlantic Atl3 region during the subsequent JJA, which then gave rise to the warm (cold) Niño3.4 SST anomalies in the following winter. As a result, a positive correlation is established between MAN NTA SST and DJF Niño3.4 SST during 1950s and 60s with the Atlantic Nino events.

Since the positive correlation between NTA and Niño3.4 in Fig. 1b is not statistically significant and also the focus of this manuscript is on explaining the increasing QB tendency of ENSO in recent decades, we have decided not to mention this speculation in the revised manuscript.

2. The coupling mechanism during JJA and thereafter looks pretty much like happening through the Walker Circulation, as suggested in Rodríguez-Fonseca et al. (2009) and Ding et al. (2011).

REPLY: Thanks for pointing out these two studies. We cited them in the introduction of the revised manuscript (Lines 83 and 168).

The Walker circulation mechanism suggested by Rodríguez-Fonseca et al. (2009) and Ding et al. (2012) was put forth primarily to explain how the equatorial Atlantic SST anomalies (Atlantic Niño) can influence equatorial Pacific SST anomalies. The ITCZ subtropical teleconnection mechanism we propose in this manuscript is more related to the NTA SST anomalies. Also, the JJA Atlantic Niño may not contribute substantially to the biennial variability in ENSO due to the weak biennial relationship between ENSO and the JJA Atlantic Niño.

In the revised manuscript, we also add statements mentioning this Walker circulation mechanism (Lines 166-170).

3. Change the color bar in Suppl. Fig. 2 for OLR and precipitation. Enhanced rainfall should be shown in blue.

REPLY: Done (See Supplementary Fig. 5c in the revised manuscript). Thanks for the suggestion.

Point-by-Point Replies to Reviewer #2

We thank the reviewer for the constructive and helpful comments. Our point-by-point replies are as follows: (The line numbers mentioned here are based on the track-change version of the manuscript)

=====

This study presents an Atlantic capacitor effect mechanism to enhance the biennial variability of the ENSO. The proposed mechanism can be summarized as follows; The El Niño tends to warm the SST over the NTA region with several months lag about one season, then, the NTA warming during boreal spring season can lead the La Niña during the subsequent winter season. This mechanism is enhanced especially during recent decades with aids of the positive phase of the AMO and global warming trend.

The proposed mechanism is plausible and convincing, however, in my opinion, it is not fully validated. One of my main concerns is that the most of results in this study can be shown even though biennial tendency of the ENSO can be excited by other factors except for the NTA. For example, both the biennial tendency of the NTA SST and ENSO would be enhanced if the biennial tendency of the ENSO can be led by the Indian Ocean-ENSO coupling. To address this concern, authors should perform the additional analysis like partial regression to exclude the factors that can lead the biennial tendency of the ENSO, and the model experiments can be another option to clarify their mechanism.

Therefore, I recommend this article to be published after the major revisions.

REPLY: Thanks for giving us the chance to revise the manuscript.

Major Comments:

1. Authors argue that the NTA can play a critical role to the biennial tendency of the ENSO after early 1990s. The mechanism is promising, but I feel the supporting materials are not clear. For example, as the Indian Ocean warming can lead the biennial tendency of the ENSO as authors mentioned in the introduction, it is worthwhile to check whether their relationship is still valid after removing the impact of the Indian Ocean warming. Also, the foot-printing mechanism that infers the interaction between the tropical and mid-latitude Pacific is worthwhile to be removed. In addition, authors show their proposed mechanism is valid for individual cases in more detail as shown in Supplementary Fig. 4.

REPLY: Following Reviewer's suggestion, we repeated the correlation analysis presented in Figure 1 after removing the influence of the Indian Ocean warming (represented by the Indian Ocean Basin index; SST anomaly over 20°N-20°S and 40°E-110°E). The new results (shown in Figure B1) are similar to those reported in Figure 1b, both of which show significant negative correlations between the MAM NTA SST and the following DJF Niño3.4 indices after the early-1990s. This new analysis indicates that the increased biennial ENSO tendency after the early 1990s is very likely related to the increased connection between Atlantic and Pacific SST anomalies and probably not due to the influences of the Indian Ocean. Text discussing this new analysis had been added in the revised manuscript (Lines 131-136).

Figure B1. The 21-year sliding partial correlation coefficients between MAM NTA SST and DJF Niño3.4 with the influences of the MAM Indian Ocean Basin index removed. a. Correlation between the preceding DJF Niño3.4 and MAM NTA SST. b. Correlation between MAM NTA SST and subsequent DJF Niño3.4. Red dots denote the correlation coefficients that are statistically significant at the 95% level.

As for the comment on “the foot-printing mechanism that infers the interaction between the tropical and mid-latitude Pacific”, the subtropical teleconnection along the Pacific Intertropical Convergence Zone (ITCZ) is necessary to enable the NTA to exert an influence on the tropical Pacific. Therefore, we think it is important to keep these statements (Lines 161-166) in the manuscript.

Lastly, we would like to thank the reviewer for her/his comment expressing appreciation (and encouragement) for our case study on the 2009-10 El Niño event, where we were able to verify the proposed Pacific-Atlantic coupling mechanism.

2. The coupled model experiments with and without the role of the NTA SST variability are worthwhile to be considered. As mentioned in the comment #1, the analysis using the observational data has a limitation to exclude various factors to lead the biennial tendency of the ENSO. The model experiment clarifies their mechanism is critical to the enhanced biennial tendency during recent decades. Or, at least authors should analyze the long-term integration of the coupled models whether their mechanism is supported in the modeling framework.

REPLY: Following Reviwer's suggestion, we have performed three ensemble experiments with a coupled model (the NCAR CESM 1.1.2). The experiments are designed to examine the influences of the NTA SST (the Atlantic capacitor effect) on Atlantic-Pacific lead-lagged correlations and ENSO periodicity.

The three coupled atmosphere-ocean general circulation model (CGCM) experiments include a control (CTRL) run, a positive AMO run, and a negative AMO run. For the AMO runs, surface heat flux anomalies (i.e., AMO-regressed SST anomalies multiplied by a factor of 100) were prescribed over the North Atlantic to force the model to produce positive or negative AMO anomalies. This modeling methodology is adapted from Zhang and Delworth (2007). The simulated AMO index is +0.4°C in the positive AMO run and -0.4°C in the negative run (compared with that in the CTRL run), which is about two times of the observed amplitude of AMO index (Sutton and Hodson, 2005). A total of 100 years of integrations from the CTRL run and 96 years from each of the two AMO runs were used in the analysis.

We first examine the “charging” and “discharging” processes in the model experiments by comparing the correlations between the NTA SST and Niño3.4 indices in the positive and negative AMO runs (Figures B2a, b). In the positive-AMO ensemble run, a significant positive correlation was found between the preceding DJF Niño3.4 and MAM NTA SST that represents the “charging” process of the capacitor effect, while a significant negative correlation was found between the MAM NTA SST and the following DJF Niño3.4 (Figures B2a, b) that represents the “discharging” process of the capacitor effect. In contrast, in the negative-AMO ensemble run, the significant positive correlation can only be but not the negative correlation between spring NTA SST and the following winter Niño3.4. These modeling results confirm the observational finding that the feedback from the Atlantic NTA region to the Pacific ENSO is stronger in the positive phase of the AMO than in the negative phase of the AMO.

We also examine the ENSO periodicity in the model experiments (Figures B2c). The CTRL run indicates that the NCAR CESM model is dominated by a 6-year ENSO. The biennial periodicity of the simulated ENSO in the model is very weak. The CESM periodicity is likely dictated by Pacific air-sea interaction processes that give rise to the low-frequency (i.e. 4-6 years) ENSO. In the positive-AMO run, the Atlantic capacitor effect provides an additional coupling dynamics that tends to give rise to a QB ENSO. A nonlinear combinations of these ENSO periodicities can give rise to ENSO periodicities near 4 years $[(6\text{year}+2\text{year})/2=4\text{year}]$. It is encouraging to see that the dominant ENSO period in the positive-AMO run splits into one centered close to 6 years and another centered at 4 years. No such splitting of the ENSO periodicity was found in the negative-AMO run, where the ENSO periodicity is still dominated by a single peak around 5 years- consistent with the lack of an Atlantic capacitor effect.

These modeling experiments support our observational findings that the Atlantic capacitor effects work more efficiently in the positive AMO phase, which then contributes to the enhanced biennial variability in ENSO during recent decades. These new results are added in Lines 225-251 of the revised manuscript.

Figure B2 Changes in the effectiveness of the Atlantic capacitor effect in numerical experiments with different background surface temperatures of the tropical Atlantic Ocean in positive and negative AMO phases. a) The “charging” process represented by the correlation coefficients between the winter Niño3.4 index and the following spring NTA index in the positive AMO run and the negative AMO run. The dashed line denotes the 95% confidence interval. b) The “discharging” process represented by the correlation coefficients between the spring NTA index and the subsequent winter Niño3.4 index. The previous winter Niño3.4 regressions have been removed from NTA index before calculating the coefficients. c) Power spectra of the winter Niño3.4 indices in the CTRL, positive and negative AMO runs. The dashed line denotes the 95% confidence interval against red noise. d) As in (c) but for the spring NTA indices.

3. Authors show using the moving correlation that the inverse relationship between the NTA warming and the subsequent ENSO is gradually increasing during recent decades. They suspected the climatological SST warming over the Atlantic related to the AMO and the global

warming is responsible for this enhanced relationship. However, the SST related to the AMO does not show any linear-trend, which might imply that the global warming might strongly affect to the enhanced NTA-subsequent ENSO relationship during recent decade. Authors need to clarify how the AMO can play a role to the enhanced NTA-subsequent ENSO relationship.

REPLY: We did mention in the original manuscript (Lines 49-53) that global warming, together with the positive phase of the AMO, contribute to the enhanced relationship between the NTA warming and the subsequent ENSO. The possible interference from global warming may be a reason why the previous positive phase of the AMO (during 1930s-1960s) was not accompanied by enhanced Atlantic-Pacific interactions. As shown in Figure 1b, the correlation between the MAM NTS index and the following DJF Niño3.4 index is significant and negative during the recent positive AMO period (after early 1990s) but weak and not significant during the previous positive AMO phase (before 1960s). This information is added in the revised manuscript (Lines 215-221).

Minor comments:

1. page3 : There are many previous literatures to examine the inter-decadal variation of the ENSO frequency (An and Wang, 2000; Wang et al., 2008).

References:

An, S. I., & Wang, B. (2000). Interdecadal change of the structure of the ENSO Mode and its impact on the ENSO frequency. *Journal of Climate*, 13(12), 2044-2055.

Wang, B., Yang, J., Zhou, T., & Wang, B. (2008). Interdecadal changes in the major modes of Asian-Australian monsoon variability: strengthening relationship with ENSO since the late 1970s. *Journal of Climate*, 21(8), 1771-1789.

REPLY: Thanks for the suggestion. These two references are cited in the revised manuscript (Line 75).

2. page3, line87: There is another paper from Ham et al. (2013) about the role of the NTA warming.

Reference:

Ham, Y. G., Kug, J. S., & Park, J. Y. (2013). Two distinct roles of Atlantic SSTs in ENSO variability: north tropical Atlantic SST and Atlantic Niño. *Geophysical Research Letters*, 40(15), 4012-4017.

REPLY: Thanks for the comment. The paper from Ham et al. (2013) concerning the role of the NTA warming is mentioned in the revised manuscript (Line 88).

3. page157-178: The linkage between the Atlantic and the Pacific through two summer Pacific high-pressure system (WPSH, and NPSH) needs to be elaborated for the easy understanding of the proposed mechanism. Does the weak biennial tendency of the NPSH in recent decades imply that the role of NPSH is not critical on the proposed mechanism?

REPLY: As suggested in the study of Yun et al. (2015), the biennial- ENSO is associated with the in-phase pattern of the WPSH and NPSH, while the low-frequency-type ENSO (a typical period of 3–7 years) is associated with the out-of-phase pattern. Here we note that the in-phase pattern between WPSH and NPSH contributes to the biennial tendency of the ENSO, as suggested by Yun et al. (2015). Based on our results, we propose that the NTA warming can contribute to establish the in-phase pattern between WPSH and NPSH after the early-1990s, by initiating subtropical teleconnection along the ITCZ and exerting mutual influences on the WPSH and NPSH.

Our point here is that it is the in-phase pattern of WPSH and NPSH is critical to the biennial tendency of ENSO. Since the NPSH is involved in the discharging process of the Atlantic capacitor effect, it plays an important role in enabling the NTA to enhance the biennial ENSO tendency. We have re-worded our statements in the revised manuscript (Lines 180-187) to more clearly elaborate these points.

Point-by-Point Replies to Reviewer #3

We thank the reviewer for the constructive and helpful comments. Our point-by-point replies are as follows: (The line numbers mentioned here are based on the track-change version of the manuscript)

=====

In this manuscript, the authors investigated the atmospheric bridges between the tropical Pacific and the tropical Atlantic using statistical analyses. They showed that the North Tropical Atlantic plays an important role in the transition between El Niño and La Niña. This mediating effect becomes more important since the early 1990s. I think this work is very interesting. In particular, Fig1b shows a beautiful relationship, which may contribute to the predictability of El Niño events.

Nonetheless, I do have several concerns and suggestions especially about the evidences of this study, the details of which are listed as follows:

1. Figure 1b is the key figure of this paper (it shows a very beautiful result, indicating that the role of Atlantic has changed during the past 50 years). Here I suggest the authors better clarify their method: How did they calculate the sliding correlation coefficients and the significance level? Whether or not did the authors remove the linear trend within each sliding window? How did the authors estimate the degree of freedom when estimating the significance level (Considering that the 95% confidence level is a constant, I assume the authors fix the degree of freedom according to the sample size, however here I suggest them to follow the methods introduced by Bretherton et al. 1999). Clarification of the method may help strengthen their most important result.

Reference

Bretherton, C. S., Widmann, M., Dymnikov, V. P., Wallace, J. M. & Blade', I. The effective number of spatial degrees of freedom of a time-varying field. *J. Clim.* 12, 1990–2009 (1999).

REPLY: In the original presentation of Figure 1b, we removed the linear trend over the entire analysis period (1948-2015) from the time series. Here, we have repeated the analysis following Reviewer's suggestion of removing the linear trend within each sliding window (Figure C1). There are no substantial differences between Figure C1 and the original Figure 1. In Figure C1, significant negative correlations can still be found between the MAM NTA and subsequent DJF Niño34 after the early 1990s (around 1994), which coincides well with the time when the AMO phase changes from negative to positive.

In our original manuscript, the number of degrees of freedom for the significance tests was determined based on the sample sizes of the time series. Following Reviewer's suggestion, we have redone the significance tests in the revised manuscript using the method of *Bretherton et al.* (1999) to determine the number of degrees of freedom. The revised significance tests do not alter the analyses, discussions, and conclusions of the manuscript. Significant negative correlations between the MAM NTA and subsequent DJF Niño34 can still be found since the early-1990s (see Fig. C1). Revises have been made for Figure 1 in the revised manuscript.

Figure C1. The same as Figure 1b except that the linear trend within each sliding window has been removed. The red dots represent correlations that are significant at the 95% confidence level.

2. Similarly, in Figure3, did the authors remove the linear trend of each period before performing the regression analyses? Please clarify.

REPLY: In Figure3, the linear trend during each period was removed before performing the regression analyses.

3. In this paper, the authors did not provide enough evidence to show the causality and the mechanism of the teleconnection between the Atlantic and the Pacific Ocean. I suggest the authors perform numerical experiments to directly show that the Atlantic “capacitor” effect can indeed change according to the background temperature of the tropical Atlantic Ocean.

REPLY: Following Reviewer’s suggestion, we have performed three ensemble experiments with a coupled atmosphere-ocean model (the NCAR CESM 1.1.2). These experiments are designed to examine the influences of the NTA SST (the Atlantic capacitor effect) on Atlantic-Pacific lead-lagged correlations and ENSO periodicity.

The three coupled atmosphere-ocean general circulation model (CGCM) experiments are a control (CTRL) run, a positive AMO run, and a negative AMO run. For the AMO runs, surface heat flux anomalies (i.e., AMO-regressed SST anomalies multiplied by a factor of 100) were prescribed over the North Atlantic to force the model to produce positive or negative AMO anomalies. This modeling methodology is adapted from Zhang and Delworth (2007). The simulated AMO index is +0.4°C in the positive AMO run and -0.4°C in the negative run (compared with that in the CTRL run), which is about two times of the observed amplitude of the AMO index (Sutton and Hodson, 2005). A total of 100 years of integrations from the CTRL run

and 96 years from each of the two AMO runs were used in the analysis.

We first examine the “charging” and “discharging” processes in the model experiments by comparing the correlations between the NTA SST and the Niño3.4 index in the positive and negative AMO runs (Figures C2a, b). In the positive-AMO ensemble run, a significant positive correlation was found between the preceding DJF Niño3.4 and the MAM NTA SST that represents the “charging” process of the capacitor effect, while a significant negative correlation was found between the MAM NTA SST and the following DJF Niño3.4 indices (Figures C2a, b) that represents the “discharging” process of the capacitor effect. In contrast, in the negative-AMO ensemble run, only the positive correlation was found but not the negative correlation between spring NTA SST and the following winter Niño3.4. These modeling results confirm the observational finding that the feedback from the Atlantic NTA region to the Pacific ENSO is stronger during the positive phase of the AMO than during the negative phase of the AMO.

Figure C2. Changes in the effectiveness of the Atlantic capacitor effect in numerical experiments with different background surface temperatures of the tropical Atlantic Ocean in positive and negative AMO phases. a) The “charging” process represented by the correlation coefficients between the winter Niño3.4 index and the following spring NTA index in the positive AMO run and the negative AMO run. The dashed line denotes the 95% confidence interval. b) The “discharging” process represented by the correlation coefficients between the spring NTA index and the subsequent winter Niño3.4 index. The previous winter Niño3.4 regressions have been removed from NTA index before calculating the coefficients. c) Power spectra of the winter Niño3.4 indices in the CTRL, positive and negative AMO runs. The dashed line denotes the 95% confidence interval against red noise. d) As in (c) but for the spring NTA indices

We also examine the ENSO periodicity in the model experiments (Figures C2c). The CTRL run indicates that the NCAR CESM model is dominated by a 6-year ENSO. The biennial periodicity of the simulated ENSO is very weak. The CESM model is likely dictated by the Pacific air-sea interaction processes that give rise to low-frequency (i.e. 4-6 years) ENSO. In the positive-AMO run, the Atlantic capacitor effect provides an additional coupling dynamics that tends to give rise to QB ENSO. A nonlinear combination of these ENSO periodicities can give rise to ENSO periodicities near 4 years $[(6\text{year}+2\text{year})/2=4\text{year}]$. It is encouraging to see that the dominant ENSO period in the positive-AMO run splits into one centered close to 6 years and the other centered at 4 years. No such splitting of the ENSO periodicity was found in the negative-AMO run, where the ENSO periodicity is still dominated by a single peak around 5 years-consistent with the lack of the Atlantic capacitor effect.

These modeling experiments supported our observational findings that the Atlantic capacitor effect can work more efficiently during the positive AMO phase, which then contributes to the enhanced biennial variability in ENSO. These new results are added in Lines 225-251 of the revised manuscript.

4. Another question that I have is why the authors chose MAM NTA temperature as the key indicator (e.g. in Figure1 and Figure2)? I understand that the MAM NTA SST may have the strongest correlation with the preceding DJF Nino SST. However, it will be better for the authors if the rationale is further explained and this relationship is further clarified.

In addition, the correlation between the tropical Atlantic SST and the subsequent Nino 3.4 SST may depend on the seasons selected (of both Atlantic SST and Nino SST). Some additional statistically analysis would be nice to show why MAM NTA SST has been selected as the indicator.

REPLY: Previous studies have shown that tropical Atlantic warming occurs 4-5 months after the mature phase of Pacific warm events (e.g., Enfield and Mayer, 1997; Klein et al., 1999; Wang, 2002). As the mature phase of these Pacific warming events associated with El Nino generally occur generally during the boreal winter (DJF) season, the NTA warming generally peaks during boreal spring (MAM). We also perform additional analyses on the relationship between the DJF Nino 3.4 SST index and the NTA SST index in each calendar month from the ENSO developing year to the ENSO decaying year for periods P1 and P2. As shown in Figure C3, the NTA SST index in the boreal spring of the developing and/or decaying years shows the strongest correlations with the DJF Nino3.4 index during both periods (Fig. C3). Apparently, spring NTA SST anomalies have the strongest correlations with the DJF Nino 3.4 SST (the peak season of the ENSO). Therefore, we choose the boreal spring (MAM) NTA SST as the key Atlantic indicator in the analyses.

We add the following information in the revised manuscript (Lines 104-107) to elaborate why the MAM NTA SST anomalies are selected for the correlation analyses: “The DJF value of the Niño3.4 index is selected to represent the ENSO intensity during its typical peak season, while MAM is the season for which the NTA SST anomalies show the strongest correlation with the ENSO SST anomalies (Supplementary Figure 1).”

Figure C3. Correlations between the D(-1)JF(0)Nino 3.4 SST index and the NTA SST index in each calendar month during the ENSO developing year (Year(-1)) to the ENSO decaying year (Year(0)) for periods P2(a) and P1(b). The red dots represent correlations that are significant at the 95% confidence level.

5. Another concern that I have is whether the Pacific climate becomes more sensitive to the Atlantic SST since 1990s, or it is the warming trend in Atlantic that favors a La Niña mode in the Pacific Ocean, considering that there is a long term warming trend over the tropical Atlantic Ocean? This is another reason that I suggest the authors to perform some numerical simulations to directly show the causality of this relationship.

REPLY: The reviewer makes a good point. The recent Pacific cooling induced by the Atlantic warming trend (see Li et al. 2016) is known to begin in the late 1990s. However, our results indicate that the enhanced Pacific response to NTA SSTs began in the early 1990s, which coincides more closely with the time of the AMO phase transition than with that of the PDO phase transition. Additionally, the results of the ensemble CGCM experiments also support our suggestion that the enhanced Atlantic “capacitor” effect is likely due to the positive AMO phase.

Reviewers' comments:

Reviewer #1 (Remarks to the Author):

I am happy with the revisions and recommend acceptance of the manuscript.

Reviewer #2 (Remarks to the Author):

All the concerns addressed in a previous round of the review is well addressed in a revised manuscript. Therefore, I recommend to publish this article to Nature Communications as its present form.

Reviewer #3 (Remarks to the Author):

I thank the authors for their detailed responses to my previous comments. Most of the comments have been well responded. The manuscript has been dramatically improved. In particular, the coupled model simulations provided additional evidences to support their conclusion. I only have several minor comments as follows:

1. After showing the relationship, the authors provide a short description of the potential mechanisms (line 156-170). It will be better to have a more detailed discussion about these mechanisms. Especially, how the NTA warming impact on the Walker circulation.
2. AMO has a phase change during early 1980s. However, the correlation between NTA and equatorial Pacific kept decreasing since 1950s (Fig. 3b). The authors need to provide a discussion about this difference.
3. Rather than citing references, it will be necessary for the authors to introduce the model setting in detail: how did they force the Atlantic Ocean? Did they introduce additional heat flux? If so, will this additional flux impact on the relationship between Atlantic and Pacific, and will this influence be realistic? It will be necessary for the authors to better introduce the methods, and then add a discussion in their manuscript.

Point-by-Point Replies to Reviewers

We thank the reviewers for their constructive and helpful comments. Our point-by-point replies are as follows: (Note that the line numbers mentioned here are based on the track-change version of the manuscript)

Reviewer #1 (Remarks to the Author):

I am happy with the revisions and recommend acceptance of the manuscript.

Reviewer #2 (Remarks to the Author):

All the concerns addressed in a previous round of the review is well addressed in a revised manuscript. Therefore, I recommend to publish this article to Nature Communications as its present form.

Reviewer #3 (Remarks to the Author):

I thank the authors for their detailed responses to my previous comments. Most of the comments have been well responded. The manuscript has been dramatically improved. In particular, the coupled model simulations provided additional evidences to support their conclusion.

I only have several minor comments as follows:

1. After showing the relationship, the authors provide a short description of the potential mechanisms (line 156-170). It will be better to have a more detailed discussion about these mechanisms. Especially, how the NTA warming impact on the Walker circulation.

Reply: The potential mechanism by which the NTA SST can trigger ENSO events is characterized by a subtropical teleconnection along the ITCZ. This mechanism has been verified and described in detail in Ham et al. (2013). In our study, we demonstrate that this mechanism is effective only after the early 1990s, but not before. We proposed that the positive phase of the AMO and the global warming trend have provided more favorable conditions after the early 1990s for the mechanism to work more effectively, giving rise to a stronger feedback from the NTA SST to ENSO events. The following statements have been added in Lines 158-163 of the newly-revised manuscript: “The inter-basin interaction mechanism of Ham *et al.*²⁰ suggests that positive NTA SST anomalies intensify the Atlantic ITCZ to excite a Gill-type response³³ over the Eastern Pacific, which then triggers subtropical Pacific air-sea coupled interactions to give rise to negative SST anomalies in the tropical central Pacific a few months later. In this study, we show that this mechanism is at work only after the early 1990s but not before.”

The Walker circulation mechanism suggested by Rodríguez-Fonseca et al. (2009) and Ding et al. (2012) was put forth primarily to explain how equatorial Atlantic SST anomalies (Atlantic Niño) can influence equatorial Pacific SST anomalies. The ITCZ subtropical teleconnection mechanism we propose in this manuscript is more related to the NTA SST anomalies. We have already stated in the previously-revised manuscript that “It should be noted that Atlantic SST anomalies can also impact Pacific SST anomalies in the following seasons by modulating the Walker circulation^{18,19}, but this mechanism primarily involves SST anomalies in the equatorial Atlantic (such as those associated with Atlantic Niño events) but not the anomalies in the NTA region” (Lines 171-174 in the newly-revised manuscript).

2. *AMO has a phase change during early 1980s. However, the correlation between NTA and equatorial Pacific kept decreasing since 1950s (Fig. 3b). The authors need to provide a discussion about this difference.*

Reply: Fig.1b shows that the correlation coefficient between NTA and equatorial Pacific SST anomalies has changed from insignificant positive values in the 1950s to significant negative values after the early 1990s. This decreasing trend of the values of the sliding correlations may be related to the global warming trend (see Supplementary Figure 6a). The positive phase of the AMO after the early 1990s added an additional warming to the Atlantic Ocean to boost the correlations above the 95% significance level. We have added the following statement in Lines 228-234 to note this possibility: “It is noted that the correlation coefficient between NTA and equatorial Pacific SST anomalies has changed from insignificant positive values in the 1950s to significant negative values after the early 1990s. This decreasing trend of the sliding correlations may be related to the global warming trend (see Supplementary Figure 6a). The positive phase of the AMO after the early 1990s added an additional warming in the Atlantic Ocean to boost the correlations above the 95% significance level”.

3. *Rather than citing references, it will be necessary for the authors to introduce the model setting in detail: how did they force the Atlantic Ocean? Did they introduce additional heat flux? If so, will this additional flux impact on the relationship between Atlantic and Pacific, and will this influence be realistic? It will be necessary for the authors to better introduce the methods, and then add a discussion in their manuscript.*

Reply: For the AMO runs, surface heat flux anomalies (i.e., the AMO-regressed SST anomalies multiplied by a factor of 100) are prescribed over the North Atlantic to force the model to produce positive or negative AMO anomalies. This modeling methodology is adapted from *Zhang and Delworth*. This information was mentioned in the previously-revised manuscript under “Methods” (see Lines 409-412). The prescribed heat flux was added to the surface heat flux calculated by the atmospheric model component of the coupled model when it was passed to the oceanic model component of the coupled model. The added heat flux was only used to force the oceanic model component. Thus, the additional heat flux we prescribed does not interfere with the surface heat flux anomalies that drive the atmospheric response to the NTA SST anomalies. The following statements have been added in the newly-revised manuscript (Lines 412-414) to clarify this point: “This prescribed heat flux was added only to force the oceanic model component and thus does not interfere with the surface heat flux anomalies that drive the atmospheric response to the NTA SST anomalies”.

REVIEWERS' COMMENTS:

Reviewer #3 (Remarks to the Author):

All the concerns addressed in previous reviews are well addressed. I recommend acceptance of the manuscript and congratulate the authors for their nice work!